# Evaluation of PSA and PSA Density in a Multiparametric Magnetic Resonance Imaging-Directed Diagnostic Pathway for Suspected Prostate Cancer: The INNOVATE Trial

**DOI:** 10.3390/cancers13081985

**Published:** 2021-04-20

**Authors:** Hayley Pye, Saurabh Singh, Joseph M. Norris, Lina M. Carmona Echeverria, Vasilis Stavrinides, Alistair Grey, Eoin Dinneen, Elly Pilavachi, Joey Clemente, Susan Heavey, Urszula Stopka-Farooqui, Benjamin S. Simpson, Elisenda Bonet-Carne, Dominic Patel, Peter Barker, Keith Burling, Nicola Stevens, Tony Ng, Eleftheria Panagiotaki, David Hawkes, Daniel C. Alexander, Manuel Rodriguez-Justo, Aiman Haider, Alex Freeman, Alex Kirkham, David Atkinson, Clare Allen, Greg Shaw, Teresita Beeston, Mrishta Brizmohun Appayya, Arash Latifoltojar, Edward W. Johnston, Mark Emberton, Caroline M. Moore, Hashim U. Ahmed, Shonit Punwani, Hayley C. Whitaker

**Affiliations:** 1Division of Surgery & Interventional Science, University College London, London WC1E 6BT, UK; joseph.norris@ucl.ac.uk (J.M.N.); linacarmona@nhs.net (L.M.C.E.); v.stavrinides@ucl.ac.uk (V.S.); s.heavey@ucl.ac.uk (S.H.); u.stopka-farooqui@ucl.ac.uk (U.S.-F.); b.simpson@ucl.ac.uk (B.S.S.); m.emberton@ucl.ac.uk (M.E.); caroline.moore@ucl.ac.uk (C.M.M.); hayley.whitaker@ucl.ac.uk (H.C.W.); 2Centre for Medical Imaging, University College London, London WC1E 6BT, UK; saurabh.singh1@nhs.net (S.S.); elly.pilavachi@nhs.net (E.P.); j.clemente@nhs.net (J.C.); nicola.showell1@nhs.net (N.S.); d.atkinson@ucl.ac.uk (D.A.); teresita.beeston@nhs.net (T.B.); mrishta@gmail.com (M.B.A.); a.latifoltojar@nhs.net (A.L.); edward.johnston@nhs.net (E.W.J.); s.punwani@ucl.ac.uk (S.P.); 3Department of Radiology, University College London Hospitals NHS Foundation Trust, London WC1H 8NJ, UK; alex.kirkham@uclh.nhs.uk (A.K.); clare.allen@uclh.nhs.uk (C.A.); 4Department of Urology, University College London Hospitals NHS Foundation Trust, London WC1H 8NJ, UK; alistair.grey@nhs.net (A.G.); eoin.dinneen@nhs.net (E.D.); gregshaw@nhs.net (G.S.); 5Department of Urology, Barts Health, NHS Foundation Trust, London EC1A 7BE, UK; 6Centre for Medical Image Computing, Department of Computer Science, University College London, London WC1E 6BT, UK; e.bonet-carne@ucl.ac.uk (E.B.-C.); panagio@cs.ucl.ac.uk (E.P.); d.alexander@ucl.ac.uk (D.C.A.); 7Department of Pathology, University College London Hospitals NHS Foundation Trust, London WC1H 8NJ, UK; dominic.patel.11@ucl.ac.uk (D.P.); m.rodriguez-justo@ucl.ac.uk (M.R.-J.); Aiman.Haider@uclh.nhs.uk (A.H.); alex.freeman2@nhs.net (A.F.); 8Department of Clinical Biochemistry, Addenbrookes Hospital NHS Foundation Trust, Cambridge CB2 0QQ, UK; peter.barker@addenbrookes.nhs.uk (P.B.); kab45@cam.ac.uk (K.B.); 9Molecular Oncology Group, University College London, London WC1E 6BT, UK; tony.ng@kcl.ac.uk; 10Department of Medical Physics and Bioengineering, University College London, London WC1E 6BT, UK; d.hawkes@ucl.ac.uk; 11Department of Radiology, Royal Marsden Hospital, London SW3 6JJ, UK; 12Imperial Urology, Imperial College Healthcare NHS Trust, London W2 1NY, UK; hashim.ahmed@imperial.ac.uk; 13Imperial Prostate, Division of Surgery, Department of Surgery & Cancer, Faculty of Medicine, Imperial College London, London SW7 2AZ, UK

**Keywords:** biomarkers, INNOVATE, multiparametric MRI, prostate cancer, diagnosis, PSA density

## Abstract

**Simple Summary:**

As we start to see the increased, routine, use of multiparametric magnetic resonance imaging (mpMRI) to inform prostate cancer diagnosis and image-guided biopsy, data collected from University College London Hospital describes the expected diagnostic outcomes of PSA and PSA Density in this pathway as well as the potential space remaining for novel biomarkers to further improve the pathway.

**Abstract:**

*Objectives*: To assess the clinical outcomes of mpMRI before biopsy and evaluate the space remaining for novel biomarkers. *Methods:* The INNOVATE study was set up to evaluate the validity of novel fluidic biomarkers in men with suspected prostate cancer who undergo pre-biopsy mpMRI. We report the characteristics of this clinical cohort, the distribution of clinical serum biomarkers, PSA and PSA density (PSAD), and compare the mpMRI Likert scoring system to the Prostate Imaging–Reporting and Data System v2.1 (PI-RADS) in men undergoing biopsy. *Results*: 340 men underwent mpMRI to evaluate suspected prostate cancer. 193/340 (57%) men had subsequent MRI-targeted prostate biopsy. Clinically significant prostate cancer (csigPCa), i.e., overall Gleason ≥ 3 + 4 of any length OR maximum cancer core length (MCCL) ≥4 mm of any grade including any 3 + 3, was found in 96/195 (49%) of biopsied patients. Median PSA (and PSAD) was 4.7 (0.20), 8.0 (0.17), and 9.7 (0.31) ng/mL (ng/mL/mL) in mpMRI scored Likert 3,4,5 respectively for men with csigPCa on biopsy. The space for novel biomarkers was shown to be within the group of men with mpMRI scored Likert3 (178/340) and 4 (70/350), in whom an additional of 40% (70/178) men with mpMRI-scored Likert3, and 37% (26/70) Likert4 could have been spared biopsy. PSAD is already considered clinically in this cohort to risk stratify patients for biopsy, despite this 67% (55/82) of men with mpMRI-scored Likert3, and 55% (36/65) Likert4, who underwent prostate biopsy had a PSAD below a clinical threshold of 0.15 (or 0.12 for men aged <50 years). Different thresholds of PSA and PSAD were assessed in mpMRI-scored Likert4 to predict csigPCa on biopsy, to achieve false negative levels of ≤5% the proportion of patients whom who test as above the threshold were unsuitably high at 86 and 92% of patients for PSAD and PSA respectively. When PSA was re tested in a sub cohort of men repeated PSAD showed its poor reproducibility with 43% (41/95) of patients being reclassified. After PI-RADS rescoring of the biopsied lesions, 66% (54/82) of the Likert3 lesions received a different PI-RADS score. *Conclusions*: The addition of simple biochemical and radiological markers (Likert and PSAD) facilitate the streamlining of the mpMRI-diagnostic pathway for suspected prostate cancer but there remains scope for improvement, in the introduction of novel biomarkers for risk assessment in Likert3 and 4 patients, future application of novel biomarkers tested in a Likert cohort would also require re-optimization around Likert3/PI-RADS2, as well as reproducibility testing.

## 1. Introduction

Prostate cancer represents a significant global healthcare challenge [1,2,3]. The traditional diagnostic approach of combining serum prostate specific antigen (PSA) testing with subsequent systematic transrectal ultrasound (TRUS)-guided biopsy is now widely accepted as being suboptimal for identifying men at highest risk [4,5]. The introduction of pre-biopsy multiparametric magnetic resonance imaging (mpMRI) has greatly enhanced the risk stratification of men with suspected prostate cancer, enabling the omission of biopsy in low-risk men, and use of image-guided biopsy in high-risk patients [4] This strategy reduces the number of prostate biopsies required, and results in a decrease in the overdiagnosis of clinically insignificant disease and an increase in the diagnosis of clinically significant disease [4,6,7,8] Almost all clinical guidelines now support pre-biopsy mpMRI in men with suspect prostate cancer; including; the National Institute for Health and Care Excellence (NICE), the European Association of Urology (EAU) and the American Urological Association (AUA) [9,10,11,12]. A common biomarker used alongside mpMRI is MRI-derived PSA density (PSAD), particularly in the case of intermediate or negative mpMRI results, where a threshold of 0.15 ng/mL/mL is commonly advocated [13,14,15]. The 2019 NICE guidelines suggest a PSA density threshold of 0.15 ng/mL/mL can be used as part of the decision to biopsy a men with raised PSA and negative MRI (Likert score of 1 or 2) [12]. The 2019 EAU guidelines also discuss the utility of PSAD to identify patients that can safely avoid biopsy in case of a negative mpMRI, but include no formal cutoff in their final recommendations [9].

The INNOVATE trial (ClinicalTrials.gov: NCT02689271; combIning advaNces in imagiNg with biOmarkers for improVed diagnosis of Aggressive prosTate cancer) is a prospective cohort study evaluating the value of additional fluidic biomarkers and a novel diffusion-weighted MRI technique to the mpMRI-directed diagnostic pathway for suspected prostate cancer [16] Here we present the clinical diagnostic outcomes of the mpMRI-directed pathway of the INNOVATE study and examine the distribution of mpMRI scores and PSA density (PSAD) in this cohort in relation to clinically significant cancer on biopsy. We evaluate what space remains for other serum and urine biomarkers, and compare the Likert and PIRADS version 2.1 scoring system in men with biopsy to better relate our findings to centers utilizing this scoring method.

## 2. Materials and Methods

### 2.1. Patient Population and Study Eligibility

The INNOVATE study protocol has been described in-depth elsewhere [16]. In brief, men referred with suspected prostate cancer between April 2016 to September 2019 underwent serum and urine collection, and pre-biopsy mpMRI (with additional VERDICT MRI (Vascular, Extracellular, and Restricted Diffusion for Cytometry in Tumours [17])), followed by MRI-targeted biopsy when indicated as part of usual clinical care. Inclusion and exclusion criteria for the full INNOVATE study are shown in (Appendix A) and published previously [16]. Initial radiological end points from the trial have also been published [18]. In line with the inclusion criteria, the pilot stage of the study included both men with and without a prior prostate cancer diagnosis, later recruitment stages omitted men with a prior positive biopsy in favor of men undergoing diagnosis. For this publication to better reflect a diagnostic cohort, the men with prior positive prostate biopsy (*n* = 42) were excluded, alongside men for whom biopsy was recommended clinically but not performed (*n* = 9), reasons for this included patient refusal/loss to clinical follow up (*n* = 4), patient choice to have the biopsy elsewhere/in a private clinic (*n* = 3) or biopsy was prevented for other clinical reasons (*n* = 2) (Appendix A).

### 2.2. Multiparametric MRI

All men underwent a clinical prostate mpMRI at UCLH on either a 1.5 T or 3.0 Tesla scanner (Achieva; Philips, Best, the Netherlands/Ingenia, Phillips/Avanto, Siemens, Erlangen, Germany) using a pelvic-phased array coil. mpMRI sequences included T1-weighted (T1W), T2-weighted (T2W), dynamic contrast enhancement (DCE) with gadolinium, diffusion-weighted imaging (DWI) and apparent diffusion coefficient (ADC) maps. Although there may be some improvements in image quality on a 3 T scanner (important for more technical research scans), both are sufficient for mpMRI diagnostic performance and so for the diagnostic endpoints included in this work. This is supported by work carried out by others and all scanners were above the minimum field strength suggested by a consensus of experts [14,19,20]. The additional VERDICT sequences were for research only and so not described herein or used as part of the clinical decision making for these patients. All mpMRI were scored at UCLH as part of standard clinical practice by experienced uro-radiologists including (AK, CA, SP) using a five-point ordinal Likert scale for the likelihood of clinically significant prostate cancer [14]. The ‘Likert score’ is a cognitive score based on radiologist experience after inspection of the prostate using different sequences and interpreted alongside clinical data available to them at referral. If multiple suspicious areas are present, these are scored separately. The Likert score reported in this analysis represents the highest score in any area of the prostate given to a patient. For this analysis, men who underwent biopsy also had their highest scoring lesion on mpMRI re-scored using the Prostate Imaging–Reporting and Data System v2.1 (PI-RADS) scoring system [10], this secondary reporting was done retrospectively and not used to influence clinical decision making. Blinding of lesion Likert score for re-reporting was not done due to the rigid nature of PI-RADS statements and because it was required to identify the right lesion for rescoring. Men who did not undergo biopsy were not re-scored with PI-RADS. This was because of ethical implications as well as a lack of histopathological outcome which would allow a meaningful comparison between the two scoring systems.

### 2.3. Biopsy

INNOVATE is a non-interventional trial and so decision to biopsy in this cohort reflects clinical practice and the current NICE guidelines, i.e., each patient received an individual decision to biopsy based on mpMRI results, PSA/PSAD level as well as other clinical information if it is available including (but not limited to); PSA velocity, DRE findings, family history, other risk factors and comorbidities as well as life expectancy. The decision is made in discussion with the patient and so patient choice is also a key factor. In this cohort prostate biopsy was performed a median of 28 days after mpMRI (IQR: 18–50). The majority (81%, 158/195) of biopsies were carried out at UCLH, these biopsies were taken transperineally and consisted of MRI-targeted deployments only (cognitive method) with 1% (2/158) of men having additional systematic sampling if indicated clinically. Reflecting real-world variation in clinical practice 19% (37/195) of men in this cohort had their care transferred to a sister hospital (Barts Health), biopsies carried out here were taken transrectally and the majority (76%, 28/37) consisted of MRI-targeted deployments (cognitive method) with additional systematic sampling (Appendix A). Clinically significant prostate cancer (csigPCa) is defined as overall Gleason ≥3 + 4 of any length OR maximum cancer core length (MCCL) ≥4 mm of any grade, concordant with definition 2 used in the PROMIS validation report [4]. These criteria were previously developed and validated for detection of Gleason score 4 or greater and cancer core lengths representative of lesions 0.5 mL or greater [21]. Data broken down by other definitions are available as Appendix A.

### 2.4. Clinical Serum PSA and PSA Density (PSAD)

INNOVATE recruited at secondary care level from a population of men who were referred with a ‘suspicion for prostate cancer,’ so accordingly 82% (283/340) of the men in this cohort had a referral PSA level above the age adjusted referral limit (>3.0 ng/mL for men aged 50–69 years old, and >5.0 ng/mL for men at or over 70 years old [22]). The clinical serum PSA measurements were taken from the clinical notes and PSA testing was carried out in primary or secondary care prior to referral or around the time of mpMRI. The median time between PSA measurements and mpMRI in this cohort was—22 days (IQR—49 to 0 days). The median time between PSA measurement and prostate biopsy (for those who had one) was 33 days (IQR 9 to 68 days). PSA density (PSAD) was calculated by dividing serum PSA (ng/mL) by mpMRI-derived prostate volume (mL) measured using the prolate ellipsoid method (width × length × height × 0.52) [14].

### 2.5. ‘Lab’ PSA Measurement

For the figure describing reproducibility of PSAD new serum samples were collected at the point of patient mpMRI for analysis of total serum PSA. Samples were taken a median of 13 days after the patient’s clinical PSA test was done and all were within 3 months (IQR = minus 39 to plus 13 days). All samples were sent for re-testing at an external clinically accredited facility; Core Biochemical Assay Laboratory (CBAL) at Cambridge University Hospitals NHS Foundation Trust. The following assay was used; Total PSA (Beckman Coulter Access Hybritech PSA (Ref 37200) using an Access 2 immunoassays analyzer (Beckman Coulter Ltd, High Wycombe, UK). Total PSA in ng/mL was measured directly in patient serum and made comparable to a standard curve. The acceptance of each batch was undertaken by running quality control samples at the beginning and end of each assay, with the analysis being accepted only when the quality controls were within the predefined limits. The clinical and the repeated lab PSA were positively correlated with an rho^2^ of 0.4 (*p* = 3 × 10^−10^). There was no correlation between the difference between the two values and time between the two tests (r^2^ of 0.006 (= 0.5) (Appendix A).

### 2.6. Statistical Analyses

All statistical analyses were performed in RStudio version 1.3.1056 (© 2009-2020 RStudio, PBC. Boston, MA, USA) within the R statistical environment v4.0.2. Comparison of median tests used include; Independent 2-group Mann-Whitney U Test (stats package v4.0.2 wilcox.test) and Kruskal-Wallis Rank Sum Test with Dunn’s Kruskal-Wallis Multiple Comparisons (stats package v4.0.2 kruskal.test and FSA package 0.8.30 dunnTest) these tests use PSA or PSAD as numeric input data and grouping via ordered factors (either MRI score or biopsy outcome). For correlation tests PSA or PSAD values were log2 transformed before correlation using Spearman’s rank correlation (two.sided, exact = FALSE) (stats package v4.0.2 cor.test).

## 3. Results

### 3.1. Clinical Outcomes

In this cohort, 340 men underwent pre-biopsy mpMRI for prostate cancer diagnosis, of these 57% (195/340) had a subsequent MRI-targeted prostate biopsy (Table 1). The proportion of men who had biopsy increased with mpMRI Likert score; 4% (2/47) of men with Likert score 2, 46% (82/177) of men with Likert score 3, 93% (65/70) of men with Likert score 4, and 100% (46/46) of men with Likert score 5 had a biopsy (Table 1).

Of the 195 men who had a prostate biopsy clinically significant prostate cancer (csigPCa) was diagnosed in 49% (96/195), rising to 56% (110/195) for cancer of any grade or size (Table 1). As a proportion of men biopsied in each Likert group, the number of men who had csigPCa increased with Likert score; 0% (0/2) of men with Likert 2, 15% (12/82) of men with Likert score 3, 60% (39/65) of men with Likert score 4, and 98% (45/46) of men with Likert score 5 (Table 1). The number of men in these groups above and below a PSAD threshold of 0.15 (or 0.12 for men aged <50) is shown in (Figure 1) and discussed in more detail for Likert 3 and 4 men below.

For the men who underwent biopsy the mpMRI lesion that scored highest was re-scored with PI-RADS retrospectively. After PI-RADS rescoring, 60% of lesions had the same score as their Likert score (117/195), of the remaining lesions 11% (21/195) had a higher PI-RADS score and 29% 57/195 had a lower PI-RADS score, no lesions underwent more than one category change up or down the scale (Table 2). After rescoring the proportion of patients in each PI-RADS group that had csigPCa on biopsy is also shown alongside Likert in (Figure 1 and Table 2). However, because Likert was used to select for biopsy in this cohort the absolute proportions for each PI-RADS group that would have been selected for biopsy or have had csigPCa on biopsy cannot be known.

Overall biopsy outcome of Likert 2 and 5 patients is shown in (Table 1 and Figure 1), however due to the high proportion of patients in these groups without and with csigPCa respectively, discussion on the space for novel serum and urine biomarkers will focus on Likert 3 and 4 men. The detail on the Likert 2 and 5 men can still be read in Appendix A.

### 3.2. Likert 3

The number of Likert 3 men who underwent subsequent biopsy in our cohort was 46% (82/177) (Table 1). PSAD was considered clinically in the decision to biopsy, despite this it is clear other clinical variables are playing a role in the decision making because 67% (55/82) of the Likert 3 men who underwent prostate biopsy had a PSAD below the threshold of 0.15 (or 0.12 for men aged <50) (Appendix A). For the men who were below this threshold, the proportion that had csigPCa on biopsy was 9% (5/55), compared to 26% (7/27) of men above the threshold (Figure 1, Appendix A). This data for other definitions of csigPCa and other PSAD thresholds is available in Appendix A.

After PI-RADS rescoring of the biopsied Likert 3 lesions 66% (54/82) of the lesions received a different level of PI-RADS score compared to Likert; with 12% (10/82) receiving a PI-RADS 4 and 54% (44/82) receiving a PI-RADS 2 (Table 2). This large number of reclassified lesions reflects that Likert 3 (as used in UCLH) is a clinical definition that encompasses more than just the prescribed mpMRI features prescribed in PI-RADS. If the Likert 3 men that received a PI-RADS 2 score had not been biopsied a further 13% (44/340) of men could have been spared biopsy, however 4.5% (2/44) of these men had csigPCa on biopsy, increasing to 14% (6/44) if cancer of any definition is used (Figure 1 and Figure 2). The two men that had csigPCa on biopsy had PSAD (PSA) values of 0.14 (6.1) 0.14 and 0.19 (4.2), one above the clinical PSAD threshold and one below. Comparatively of all the Likert 3 men with PI-RADS 2 lesions (*n* = 44), 30% (13/44) would have also had a PSAD above the threshold. The median (IQR) PSAD for these men was 0.13 (0.10–0.15), significantly higher than the PSAD median (IQR) for all Likert 2 men; 0.09 (0.06–0.12) (*p* = 0.003), suggesting either this influenced their characterization as Likert 3, or some feature of mpMRI (yet to be included in the PIRADS definition) associates with a higher PSAD (Appendix A). The full distribution of PSA and PSAD values for all men with Likert 3 can be seen graphically in (Appendix A).

### 3.3. Likert 4

In the Likert 4 group, five men did not have a biopsy (5/70), four out of these five men had very low PSAD (PSA) values as follows; 0.04 (14), 0.06 (7.54), 0.05 (4.5), 0.02 (0.68), and the remaining man was 0.12 (7.85). The median (IQR) PSAD of all Likert 4 men was 0.13 (0.10–0.23) so the majority of the men spared biopsy were outliers in the lowest quartile of PSAD values. The full distribution of PSA and PSAD values for all men with Likert 4 can be seen graphically in Appendix A. Of the remaining men who did have biopsy, 55% (36/65) of these men had PSAD values below the PSAD threshold of 0.15 (or 0.12 for men aged <50) (Appendix A). In the men below the PSAD threshold the proportion that had csigPCa on biopsy was 53% (19/36), compared to 69% (20/29) of men above the threshold (Figure 1, Appendix A). This data for other definitions of csigPCa and other PSAD threshold is available in Appendix A.

Since almost all men with Likert 4 were biopsied (65/70) we carried out confusion matrix analysis to see how good a range of PSA and PSAD thresholds would have been in this cohort to determine if Likert 4 men could avoid biopsy (Figure 2). In this context it is important to have low numbers of false negatives, to ensure every man with csigPCa gets a biopsy. However, to achieve a false negative value of 5% the false positive rate was 31%, if this threshold (0.09 ng/mL/mL) was applied to out cohort, 86% of all men in the group would be above it (Figure 2). Interestingly a similar predictive ability could be obtained with a PSA value above 4; i.e., a false negative value of 6% and false positive rate was 34% suggesting better biomarkers would be useful in this context. It should be noted that PSAD outperforms PSA at lower thresholds, i.e., with a greater false negative rate (Figure 2). Other definitions of cancer are also available in Appendix A.

When Likert 4 lesions were rescored with PI-RADS, 4.6% (3/65) were rescored as PI-RADS 3 and 17% (11/65) were rescored as PI-RADS 5. Of the 3 downgraded to PI-RADS 3, 60% (2/3) had csigPCa on biopsy. Of the 11 upgraded to PI-RADS 5 90% (10/11) had csigPCa on biopsy (Figure 1, Table 2).

### 3.4. PSA and PSAD Reproducibility

To investigate the effect of PSA reproducibility on this cohort; total serum PSA was re-tested in the first 95 patients, serum samples were collected at the point of mpMRI and sent for re-testing at an external clinically accredited facility. When the repeated PSA values were converted to PSA density (PSAD) using the prostate volume as measured on mpMRI; 57% (54/95) maintained their classification as above or below a PSAD of 0.15, but 43% (41/95) would have been reclassified (Appendix A). This demonstrates the importance of repeatability for any new biomarkers, or panels of biomarkers aiming for implementation in this space.

## 4. Conclusions

Here we present the key diagnostic outcomes of the INNOVATE trial, reflecting real-life clinical outcome measures from experienced secondary care teams using a mpMRI-directed clinical pathway for men with suspected prostate cancer. Early experience at UCLH of using pre-biopsy mpMRI and PSAD to guide the decision to perform biopsy, 43% of men in this cohort were spared biopsy. Although not directly comparable, this is in line with previously published studies that use mpMRI alone to estimate the proportion of men who can be spared biopsy; 27% of men (derived from the PROMIS study [4]) or 50% of men (3 T in bore biopsy study [24]). In our cohort, 47% of men indicated for biopsy were diagnosed with PGG 2 or higher, this is similar to the MRI-targeted biopsy arm of the PRECISION trial (38%) [8]. Any differences in patient numbers between our and historic UCLH cohorts cannot be reliably interpreted because over time UCLH has built on years of experience which has caused changes in its decision making processes and/or biopsy technique, and because all patients in this analysis were consented to a clinical trial and the recruitment or consenting process itself can self-selected for a certain group of patients.

In summary there remains space for novel serum and urine biomarkers in this new pathway. Around 35% (51/147) of men with Likert 3 or 4 disease on mpMRI were still suspicious for cancer and underwent prostate biopsy with no resulting clinically significant cancer diagnosis. The PSAD level of 0.15 ng/mL/mL used as a trigger for biopsy in men with negative MRI (Likert 1 and 2) did not perform well enough in this cohort to help risk stratify patients of other Likert scores. Post hoc testing of other thresholds in the Likert 4 subgroup shows some patients could be spared but to get sufficient sensitivity the threshold would have to be so low almost all men would still be indicated for biopsy. Novel biomarkers in this space would aim to spare men biopsy, so a low false negative rate would be important.

When biopsied lesions were rescored with PIRADS, Likert 3 lesions received the lowest number of unchanged scores, something seen previously in a different cohort from the same center by Brizmohun and colleagues [25]. This suggests any novel biomarkers developed and validated in a Likert cohort would require particular re-optimization around PI-RADS 2 and 3 before application to a PI-RADS cohort. Some significant cancers would have been missed in the PI-RADS 2 lesions that underwent biopsy in our cohort, so novel biomarkers could also have utility here. Pagniez et al., discuss the utility of PSAD for mpMRI invisible cancers like these in a recent meta-analysis [13].

Despite improvements in the standardized (and less subjective) PI-RADSv2 reporting system, Likert is still preferred in our center. This preference was supported by a panel of UK experts in 2018 [14]. Potential problems with inter-reader variability that could affect our results and/or diagnostic outcomes with Likert score are reduced by group re-assessment as part of the standard of care at our center: At UCLH patient mpMRI images are reviewed in multi-disciplinary team meetings in which clinical decisions are made and, where indicated, consensus maps drawn up for biopsy targets. The inter reader reproducibility of Likert scores and comparisons with PIRADSv2 are well detailed elsewhere [20,26,27,28,29].

Our comment of the utility provided by PSA and PSAD for prediction of clinically significant disease in specifically Likert 3 patients was limited by the unblinding of clinicians to these markers, as well the fact 55% (95/177) of men with Likert 3 disease did not undergo biopsy, and so had no confirmed histopathological outcome. Because the clinical decision to biopsy (and Likert scores) were already influenced by PSA and PSAD, any analysis of biopsy outcome is potentially subject to selection bias. However this question has been comprehensively answered by others in a study with template biopsy for all patients as the reference standard. In a *post-hoc* analysis of the PICTURE trial, men with indeterminate (i.e., Likert 3) lesions and clinically significant disease on biopsy were found to have significantly higher PSAD compared to men without significant disease (0.19 vs. 0.13; *p* = 0.004), and a threshold of >0.17 ng/mL was shown to improve prediction of significant cancer on biopsy, but 9% of patients with significant cancer would be missed [25]. This aligns in our cohort; in the biopsied Likert 3 men there was a difference in the median PSAD between men with and without csigPCa; 0.12 (IQR 0.08–0.15) and 0.20 (IQR 0.13–0.26) (*p* = 0.02) the higher level being associated with csigPCa (Appendix A).

Ideally an alternative study design in which we offered biopsy to all men after MRI would have allowed for a better ground truth and validation of our endpoints, however in the current era this would be difficult ethically to implement, as it would mean putting men at unnecessary risk of complications caused by unnecessary prostate biopsy [7]. The underlying pathology of the men who are spared biopsy in this cohort is highly likely to be exclusively clinically insignificant cancer, this is based on historical academic work from our own center, hence why it is now standard of care here to not biopsy these men [4,30]. To support this assumption, the clinical notes for all patients in the ‘no biopsy’ group were double checked to make sure these men truly were ‘no longer suspicious’ for prostate cancer after mpMRI. Any men not undergoing biopsy for another reason were removed prior to analysis (Appendix A).

The repeatability of total PSA has been studied elsewhere [31]. In the Roehrborn study retrospective analysis of clinical data from one center showed one third of the patients had a difference of greater than ± 1.0 ng/mL on repeat PSA. In our study this value was higher at 85% of patients. The additional variability is likely due to the fact samples were measured in different clinical testing facilities and most likely using different testing platforms. If an absolute threshold for clinical action is to be implemented with PSAD, the reproducibility of PSA testing needs to be addressed more fully. This problem with reproducibility would have to be something other biomarkers or larger panels of biomarkers should aim to improve on.

Other potential limitations of the analysis are based firstly on the INNOVATE trial itself, and secondly, on the post-hoc nature of our study. mpMRI was performed at a single tertiary referral centre with extensive experience with mpMRI and therefore somewhat limited external validity, and generalisability, particularly for smaller centres who are still in the early phase of mpMRI adoption. INNOVATE was primarily an observational trial in which no diagnostic or treatment decisions were made based upon novel aspects of the trial, so any discovery would need validation in future clinical studies. The inherent heterogeneity in prostate biopsy techniques (including, systematic, targeted and combination approaches) used in INNOVATE (Appendix A) mean that correlation between biochemical, radiological and pathological outcomes are imperfect, unlike other uniformly applied, rigorous biopsy approaches, such as the 5 mm transperineal mapping biopsy technique used in PROMIS [4] or PICTURE [32]. Our cohort contained 20% Transrectal (37/195) vs. 80% Transperineal (158/195) biopsies (Appendix A) and although transrectal prostate biopsy is more likely to miss csigPCa cancers the majority of the transrectal biopsies (97%) in this cohort also underwent some systematic sampling (Appendix A), so the risks of missing csigPCa are lower and so unlikely to have affected our results. Also where a non template approach is used there is potential for patients with a lower PSA density, and so a higher volume prostate on average to have lower sampling density on biopsy. However the use of extensive template biopsy for diagnosis in a clinical environment is no longer ethically appropriate and studies such as PRECISION have shown the superiority of this approach over systematic TRUS sampling [8].

The INNOVATE trial gives a clear, real-world illustration of a successful mpMRI-directed diagnostic pathway, and any samples and data acquired during this trial will provide a useful resource to help test novel biomarkers for risk stratification of men with suspected prostate cancer. Future biomarker analyses of the INNOVATE cohort will focus on validation of novel and established blood and urine fluidic biomarkers, used at the point of mpMRI and the assessment of the utility of these panels to stratify and refine the mpMRI-directed pathway for suspected prostate cancer.

## Figures and Tables

**Figure 1 cancers-13-01985-f001:**
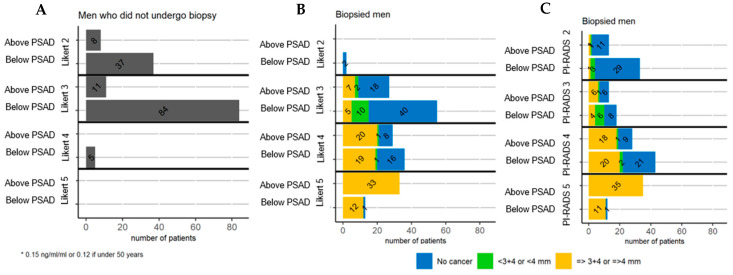
Men within the INNOVATE cohort were separated into those who did (**B**) and did not (**A**) undergo prostate biopsy after mpMRI. Within these two groups men are subdivided into 8 groups shown as bars on the Y-Axis. Groups are defined by the highest scoring lesion on mpMRI (Likert score 2, 3, 4 or 5) and if their PSA Density (PSAD) was below or above a threshold of 0.15 ng/mL/mL or 0.12 for men under 50 years old. The bar size reflects the number of men in each group, numbers are shown as text overlaying each bar. Men who did not undergo biopsy as colored Grey. When patients underwent prostate biopsy, the bars are further subdivided into 3 groups depending on the resulting overall pathology of their biopsy (Blue = No cancer of any grade, Green = clinically non-significant cancer, Yellow = clinically significant prostate cancer.). The final graph on the far right (**C**) represents the same men as detailed in (**B**) however in this graph they are grouped by the highest scoring lesion on mpMRI re-scored as PI-RADSv2. (score 2, 3, 4 or 5).

**Figure 2 cancers-13-01985-f002:**
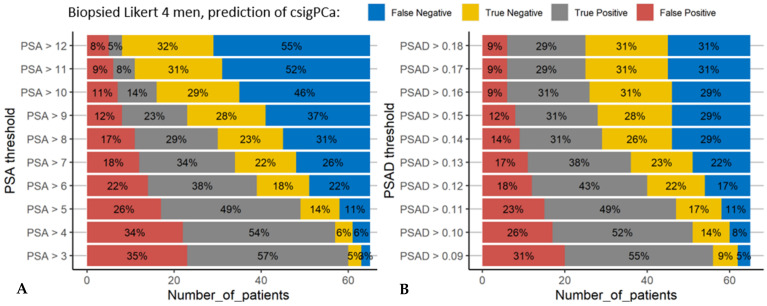
Utility of PSA and PSAD thresholds in patients scored as Likert 4. A range of thresholds of PSA (**A**) or PSAD (**B**) were used to predict the presence of clinically significant prostate cancer on biopsy (Negative = no csigPCa, Positive = yes csigPCa). Each bar represents how accurate each threshold would be if used as a clinical test to predict for csigPCa on biopsy: e.g., for the highest PSAD threshold 0.18 (top bar graph B): if a patients PSAD was <0.18 they would be ‘negative’ on the test and they would have around a 1 in 2 chance of cancer (as True Negative and False Negative are 31% and 31%), if their PSAD was >0.18 they would be ‘positive’ on the test and they would have around a 3 in 4 chance of cancer (as False Positive and True Positive are 9% and 29% respectively). In this context we potentially selecting men for biopsy so would need a very low false negative, e.g., if we use the lowest PSAD threshold of 0.09 we get a 5% false negative rate and the highest true positive rate (55%) but 86% of all men tested would get a positive test result (False Positive plus True Positive) limiting the utility of the test. This data is for Likert 4 men only.

**Table 1 cancers-13-01985-t001:** Summary demographic, clinical, radiological and pathological outcome data for all men included in the INNOVATE trial, stratified by highest Likert score. Grouped by (Table **A**) decision to biopsy and (Table **B**) presence of clinically significant prostate cancer on biopsy. Biopsy result reported as prognostic grade groups (PGG). Biopsy results are reported as prognostic grade group (PGG), endorsed and accepted by the International Society of Urological Pathology (ISUP) [23].

**(A)**
Pre-biopsy mpMRI	**Patients did NOT have a biopsy** (*n* = 145)	**Patients had a biopsy** (*n* = 195)
**Likert 2**,*n* = 45	**Likert 3**,*n* = 95	**Likert 4**,*n* = 5	**Likert 5**,*n* = 0	**Likert 2**,*n* = 2	**Likert 3**,*n* = 82	**Likert 4**,*n* = 65	**Likert 5**,*n* = 46
**Age**	66 (60, 70)	65 (58, 70)	65 (55, 74)	~	62 (61, 63)	64 (58, 68)	64 (59, 70)	68 (62, 72)
**Referral PSA (ng/mL)**	5.4(4.5, 7.4)	5.1(3.3, 6.8)	7.5(4.5, 7.8)	~	4.8(4.5, 5.1)	5.8(4.3, 8.2)	7.7(5.2, 9.9)	9.6(6.8, 19.7)
**PSA Density (ng/mL/mL)**	0.09(0.06, 0.13)	0.09(0.06, 0.12)	0.05(0.04, 0.06)	~	0.08(0.08, 0.09)	0.13(0.09, 0.16)	0.14(0.11, 0.24)	0.29(0.14, 0.59)
**mpMRI prostate volume (mL)**	65(45, 84)	50(39, 80)	90(73, 130)	~	61(52, 70)	50(34, 65)	46(32, 65)	40(30, 55)
**Biopsy Result**								
Negative Biopsy	~	~	~	~	2 (100%)	58 (71%)	24 (37%)	1 (2.2%)
PGG 1	~	~	~	~	~	15 (18%)	3 (4.6%)	~
PGG 2	~	~	~	~	~	8 (9.8%)	33 (51%)	20 (43%)
PGG 3	~	~	~	~	~	~	2 (3.1%)	17 (37%)
PGG 4	~	~	~	~	~	1 (1.2%)	2 (3.1%)	3 (6.5%)
PGG 5	~	~	~	~	~	~	1 (1.5%)	5 (11%)
**Biopsy MCCL (mm)**	~	~	~	~	~	3.0(1.0, 7.2)	7.0(4.0, 10.0)	10.0(7.0, 12.2)
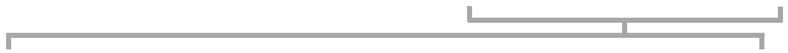
**(B)**
Pre-biopsy mpMRI	**Patients who did NOT have clinically significant prostate cancer on biopsy** (*n* = 99)	**Patients who had clinically significant prostate cancer on biopsy** (*n* = 96)
**Likert 2**,*n* = 2	**Likert 3**,*n* = 70	**Likert 4**,*n* = 26	**Likert 5**,*n* = 1	**Likert 2**,*n* = 0	**Likert 3**,*n* = 12	**Likert 4**,*n* = 39	**Likert 5**,*n* = 45
**Age**	62 (61, 63)	64 (58, 68)	64 (60, 68)	69 (69, 69)	~	58 (56, 66)	65 (59, 71)	67 (62, 72)
**Referral PSA (ng/mL)**	4.8(4.5, 5.1)	5.8(4.6, 8.4)	6.2(4.4, 10.7)	6.5(6.5, 6.5)	~	4.7(3.9, 6.6)	8.0(5.4, 9.7)	9.7(6.8, 20.4)
**PSA Density (ng/mL/mL)**	0.08(0.08, 0.09)	0.12(0.08, 0.15)	0.12(0.09, 0.16)	0.11(0.11, 0.11)	~	0.20(0.13, 0.26)	0.17(0.12, 0.26)	0.31(0.14, 0.60)
**mpMRI prostate volume (mL)**	61(52, 70)	53(36, 70)	58(39, 67)	57(57, 57)	~	30(22, 37)	38(31, 52)	40(30, 54)
**Biopsy Result**								
Negative Biopsy	2 (100%)	58 (83%)	24 (92%)	1 (100%)	~	~	~	~
PGG 1	~	12 (17%)	2 (7.7%)	~	~	3 (25%)	1 (2.6%)	~
PGG 2	~	~	~	~	~	8 (67%)	33 (85%)	20 (44%)
PGG 3	~	~	~	~	~	~	2 (5.1%)	17 (38%)
PGG 4	~	~	~	~	~	1 (8.3%)	2 (5.1%)	3 (6.7%)
PGG 5	~	~	~	~	~	~	1 (2.6%)	5 (11%)
**Biopsy MCCL (mm)**	~	1.00(1.00, 2.00)	1.50(1.25, 1.75)	~	~	7.5(5.0, 11.0)	8.0(6.0, 10.0)	10.0(7.0, 12.2)

Statistics presented: median (IQR); *n* (% of column).

**Table 2 cancers-13-01985-t002:** For the men who underwent biopsy the mpMRI lesion that scored highest was re-scored with PI-RADS retrospectively. Proportion of patients with equivalent or a different PI-RADS score stratified by Likert score and having a PSAD above or below a threshold of 0.15 (or 0.12 for men aged <50). PI-RADS was not used to select patients for biopsy.

Pre-biopsy mpMRI:	**Likert 2**, *N* = 47 *^1^*		**Likert 3**, *N* = 177 *^1^*		**Likert 4**, *N* = 70 *^1^*		**Likert 5**, *N* = 46 *^1^*	
**PSAD Threshold ***	**Below**	**Above**	**Below**	**Above**	**Below**	**Above**	**Below**	**Above**
**No Biopsy**	37	8	84	11	5	0	0	0
**No Cancer**								
Equivalent	2	0	7	6	14	8	0	0
PIRADS < Likert	0	0	27	11	1	0	1	0
PIRADS > Likert	0	0	6	1	1	0	0	0
**Insignificant cancer (≤3 + 3 or <4 mm)**								
Equivalent	0	0	6	1	1	1	0	0
PIRADS < Likert	0	0	3	1	0	0	0	0
PIRADS > Likert	0	0	1	0	0	0	0	0
**Clinically significant cancer (≥3 + 4 or ≥4 mm)**								
Equivalent	0	0	3	5	16	11	9	27
PIRADS < Likert	0	0	1	1	1	1	3	6
PIRADS > Likert	0	0	1	1	2	8	0	0

*^1^* Statistics presented: *n*. * PSAD threshold of 0.15 ng/mL/mL (or 0.12 if patient is younger than 50 yrs).

## Data Availability

The data presented in this study are available on request from the corresponding author.

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
