# Peer review of "Evaluation of PSA and PSA Density in a Multiparametric Magnetic Resonance Imaging-Directed Diagnostic Pathway for Suspected Prostate Cancer: The INNOVATE Trial"

_cancers, 2021, doi:10.3390/cancers13081985_

Round 1
Reviewer 1 Report
Congratulations to the authors on a fantastic manuscript which will be of huge interest to the field. My only comments are: 1) You seem to repeat the definition of cSig prostate cancer numerous times throughout the text which is not necessary, 2) The text for Figure 1 is unclear and difficult to read, and 3) There are a few very minor typos to correct and a reference that needs reformatting in the legend of Table 1.
Author Response
Dear Reviewer 1,
I would like to express my gratitude for the time and care taken in the review of our manuscript. My comments in reply to your comments are addressed in order below.
1) Thank you for spotting this, I have corrected this throughout and removed all surplus descriptions of our criteria for ‘clinically significant prostate cancer’, I have only left one in the abstract and one the first time it is mentioned in the main text.
2) I have re-written the text for figure one to try and make it clearer.
3) I have tried to spot the remaining typos and correct them and have addressed the reference you mentioned that needed reformatting.
Thank you for your detail review and I hope I have addressed all the problems you identified.
Hayley Pye
Reviewer 2 Report
I want to congratulate the authors for design and reporting the INNOVATE trial. This trial is of clinical and practical importance for the clinicians dealing with prostate cancer.
I enjoyed reading the results of the trial. I have the following questions:
- As stated in the manuscript, it is evident that the criteria for biopsy does not rely on PSAD, PSA, or Likert score alone. How would the authors validate/interpret the results of the study differently if all patients, regardless of Likert score, were biopsied?
- Patient in the INNOVATE trial have undergone transperineal biopsy. Transperineal biopsy is shown to increase cancer and clinically significant cancer more than transrectal biopsy. How would the authors incorporate the route of the biopsy in the interpretation of the results?
- A significant number of patients are re-staged after comparing PI-RADS and Likert scoring. Is it possible to have Likert scores re-evaluated for the biopsied lesions?
- Since the result of the study relies on the biopsy sample and there are significant number of patients in Likert 2 and 3 that were not biopsied, how would the authors interpret the validity of the biopsy and biomarkers?
- What was the inter person reliability of Likert among the experienced radiologist versus unexperienced radiologists?
- Was there a difference when the mpMRI was done with 1.5T vs 3T?
Author Response
Dear Reviewer 2,
I would like to express my gratitude for the time and care taken in the review of our manuscript. My comments in reply to your comments are addressed in order below.
- Thank you for this comment, I appreciate this is a limitation of the work, I have added some additional text in the discussion of the manuscript to discuss the approach you suggest as alternative study design (i.e. offer biopsy to all men after MRI). This would clearly allow a better ground truth and validation of our endpoints, however in the current era this would be difficult ethically to implement, as it would be putting men at unnecessary risk of complications caused by prostate biopsy (Loeb et al 2013). The underlying pathology of men who are spared biopsy in this cohort is highly likely to be exclusively clinically insignificant cancer, this is based on historical academic work from our own centre, hence why it is now standard of care here to not biopsy these men (Ahmed et al 2017, Norris et al 2020). To support this assumption the clinical notes for all patients in this cohort were also checked to ensure men in the ‘no biopsy’ group were truly not suspicious for prostate cancer after mpMRI, and any men not undergoing biopsy for another reason were removed prior to analysis. We have expanded on this in our discussion.
https://doi.org/10.1016/j.eururo.2013.05.049
https://doi.org/10.1016/j.eururo.2020.04.029
https://doi.org/10.1016/S0140-6736(16)32401-1
- As you rightly say, transrectal prostate biopsy is more likely to miss csigPCa cancers and as you saw our cohort does contain 20% Transrectal (37/195) vs 80% Transperineal (158/195) biopsies. The majority of the Transrectal biopsies (97%) also underwent systematic sampling so risks of missing csigPCa in this group are lower and so this is unlikely to have affected our results. I am aware however the differences in biopsy strategy do affect diagnostic outcome, but this is not the appropriate cohort to make any firm conclusions on biopsy strategy and missed cancer rates since this has been done well by others elsewhere in particular in the PRECISION clinical trial (Kasivisvanathan,et al 2018). We have added some additional discussion points and references surrounding this limitation in our discussion.
https://doi.org/10.1056/NEJMoa1801993
- Thank you for this suggestion, actually this was considered by the working group but unfortunately after a discussion amongst senior authors and clinical radiologists it was felt this (as well as the PIRADS scoring of the non-biopsied lesions) could be unethical as well as have low academic relevance to the cohort, considering the decision to biopsy and potential biopsy targets was already made using the Likert score, and so biopsy outcome could not be compared between the two scoring systems. In respect to re-scoring again with Likert please see below for a discussion on inter-read variability and the comparison of PI-RADSv2 and Likert. I hope the data we have included on this is still of sufficient interest to yourself and the readers of this journal.
- My interpretation of the validity of the biomarker analysis in the context of the high number of non-biopsied Likert 2 and 3 patients is as follows; I think the best use for biomarkers in the context of this secondary care setting is taking additional blood and urine at the point of mpMRI to help make the decision to biopsy, in this context knowing the Likert 2 (and some 3) have already been spared biopsy is sufficient, as it is not these men for whom the biomarker would need to be tested for ‘additional value’. There has been some discussion about also testing biomarkers in this cohort which would select patients for mpMRI (i.e. for use in centres where the capacity is not available), this could also be done as mpMRI ‘suspicion’ is a complete endpoint. Despite this, and as you say, I do appreciate it is not ideal there being an incomplete ground truth for prostate biopsy, however I do not believe we would have received ethics to do this (see above).
- We have not done any analysis on the data included in this work looking at the ‘inter person reliability of Likert’ and I appreciate this could be important as our analysis uses a more subjective Likert score rather than the structured PI-RADSv2. However in our centre patient mpMRI images are reviewed in multi-disciplinary team meetings, we are a high throughput centre and so these meetings contain multiple people familiar with the direct interpretation of uro-radiology images, images are shared and clinical decisions are discussed and where indicated, consensus maps are drawn up for biopsy targets. This means any problem with inter-reader variability that would affect diagnostic outcomes within our centre are averaged out clinically by group re-assessment. This makes it unlikely our results have been adversely affected by this. Despite improvements in the standardised PI-RADS_V2 reporting system, Likert is still preferred in our centre and a panel of experts recently recommended this system (Brizmohun Appayya et al 2018). I am aware others have looked at the ‘inter person reliability of Likert’ previously, as well as made comparisons between the PIRADSv2 vs LIKERT scoring system (stabile et al 2020, Rosenkrantz et al 2013, Khoo et al 2020, Latifoltojar et al 2019, Zawaideh et al 2020) and so further comments and references for all this have been added to the discussion for the reader’s benefit.
https://doi.org/10.1111/bju.14361
https://doi.org/10.1148/radiol.13122233
https://doi.org/10.1111/bju.14916
https://doi.org/10.1016/j.crad.2019.08.020
https://doi.org/10.1016/j.euo.2020.02.005
https:// doi. org/10. 1259/ bjr. 20200298
- We have not done any analysis on the data included in this work to look for differences when the mpMRI was done with 1.5T vs 3T. Although there may be some improvements in image quality on a 3T scanner (important for more technical research scans), both are sufficient for mpMRI diagnostic performance and so for the diagnostic endpoints included in this work. This is supported by a work carried out by others in Germany by Ullrich et al 2017 and discussed in a review by stabile et al 2020. And all scanners were above the minimum field strength suggested by a consensus of experts (Brizmohun Appayya et al 2018). A comment on this and the reference has been added to the methods.
https://doi.org/10.1016/j.ejrad.2017.02.044
https://doi.org/10.1016/j.euo.2020.02.005
https://doi.org/10.1111/bju.14361
Thank you for your thoughtful comments and I hope I have addressed all of your comments.
Hayley Pye